# Transmission Line Equipment Infrared Diagnosis Using an Improved Pulse-Coupled Neural Network

Jie Tong [1] , Xiangquan Zhang [2], Changyu Cai [1], Zhouqiang He [2], Yuanpeng Tan [1,*] and Zhao Chen [2]

1   Artificial Intelligence Application Department, China Electric Power Research Institute, Beijing 100192, China
2   State Grid Gansu Electric Power Company, Lanzhou 730070, China
*   Correspondence: tanyuanpeng@epri.sgcc.com.cn; Tel.: +86-188-1132-0266

**Abstract:** In order to detect the status of power equipment from infrared transmission line images under the spatial positioning relationship of the transmission line equipment, such as corridor, substation equipment, and facilities, this paper presents an improved PCNN model which merges an optimized parameter setting method. In this PCNN model, the original iteration mechanism is abandoned, and instead, the thresholding model is built by the maximum similarity thresholding rule. To ensure similarity during classifying neighboring neurons into cluster centers, a local clustering strategy is used for setting the linking coefficient, thus improving the efficiency of the method to detect the power equipment in infrared transmission line images. Finally, experimental results on transmission line infrared images show that the proposed method can provide the basis for the diagnosis of power equipment, preventing the casualties and property damage caused by the thermal damage of power equipment, and effectively improving the safety risk identification and operation control ability of power grid engineering.

**Keywords:** pulse-coupled neural network; image segmentation; parameter setting; similarity thresholding

## 1. Introduction

Transmission line inspection is important to ensure the normal operation of high-voltage transmission lines, which is of great significance to the sustainable supply of power and the safe and rapid economic development of power in China. In the inspection of power transmission lines infrared imaging technology, as an important tool for long-distance detection of detecting equipment faults, has the advantages of non-contact detection, high safety, high accuracy and simple operation. However, the detected fault information often was recorded by workers on site, increasing the inspection time and miss detection. Accordingly, image processing technology, such as image segmentation, has become a significant tool for automatically detecting equipment faults in power transmission lines.

A pulse-coupled neural network (PCNN) is an iterative model inspired by the observation of synchronous pulse bursts in the visual cortex of small mammals such as cats [1]. The underlying idea of this model was to bridge temporal gaps and encourage the neurons with spatial proximity and brightness similarity to pulse together and was soon recognized as having attractive application prospects in the field of image processing, including image segmentation [2]. For years, PCNN has been widely applied to segmenting a wide range of real-world images, such as infrared images [3–8], and other natural images [9–11].

In general, the performance of the PCNN model to image segmentation is heavily dependent on the values of network parameters. Among these parameters are amplitudes and decay constants in each leaky integrator, two weight matrices, the linking coefficient, etc., thus making it more flexible for adjusting the neuron's behavior in image segmentation. Additionally, the mechanism of parameter settings remains uncertain in processing a wide range of images. So, in some cases, some of these parameter values are often determined by trial and error [12–14].

During the last few years, much work has been devoted to parameter settings including simplifications. The simplified or modified model developed by Kuntimad and Ranganath [15] was regarded as a classic model, which may guarantee perfect segmentation for an image after obtaining the intensity ranges of adjacent regions. Followed by this model, Raya et al. [16] treated the linking coefficient and the primary firing threshold as global and local values, and then obtained their values directly from image statistics. To build the relationship between the dynamic property of neurons and the image characteristics, Chen [17] built some criteria to deal with the parameters of the simplified model. Besides, the methods in Refs [3,4] tried to establish a simple way to segment the infrared image. The derived conditions are so strict to treat with image characteristics [18]. Recently, efficient ways proposed in the literature [19,20] utilize local information for preserving the characteristics of the synchronous pulse inherent in PCNN. However, the final result has close ties with the clustering method.

As in the output of neurons, the PCNN model has a fundamental mechanism for obtaining the results via setting the parameter of the neural threshold. The general neural threshold inherent in PCNN increases once the neuron fires, and then it decays until the corresponding neuron fires again, thus creating dynamic changes for each neuron. Wei [12] established the threshold decay time constant with the overall features of target images according to the subjective feeling of human eyes, logarithmically related to the actual light intensity. Particularly, a strategy to linearly decrease the value of the threshold was utilized for the sake of computer processing [8,21–25]. However, the neural threshold by periodical decaying seems to not be fit for image segmentation because of its periodic repetition. Kong [6,7] had thereby entirely abandoned this mechanism, but instead, used the average gray value of a peak in the histogram as the corresponding threshold. In Ref. [26], Karvonen used a fixed classwise threshold value for each class which is given by the Expectation-Maximization method from the histogram. Later, Li [27] derived a method using the water region area in the histogram. In fact, the threshold of targets may be missing through histogram analysis. In the literature [3,4,20], image characteristics were used instead of histogram analysis, allowing the threshold to be updated adaptively. Usually, the behavior of these methods is similar to an alternative to global region growing but differs from the seeded region method with PCNN [28–30]. However, intensity-inhomogeneity in images often occurs, which makes PCNN unsuitable for image segmentation according to the above parameter setting methods. Besides, there are many model parameters, so it is not easy to build a reasonable setting method for promoting the performance of image segmentation.

In this paper, a novel strategy for image segmentation is presented by a modified version of PCNN. The fundamental idea of this method is to merge a similarity thresholding method. Then, a new relationship between the parameters and the available information such as the pulse output and image characteristics is built to enable the model to extract the target. The main contributions of this paper are related to two important methodological issues. The first one is related to a maximum thresholding rule to determine the threshold of the modified PCNN model. The second contribution concerns the regulation of the linking coefficient, activating similar neurons as much as possible to make the segmentation performance better.

The organization of the paper is as follows: in Section 2, the original PCNN and its modification are described for image segmentation. The maximum similarity thresholding rule is built and the linking coefficient is set by local clustering in Section 3. In Section 4, experimental results of the proposed PCNN on real-world transmission line infrared images are given, and the results are then compared with some widely used methods to demonstrate the performance of image segmentation; finally, conclusions are drawn in Section 5.

## 2. Pulse-Coupled Neural Network and Its Modification

### 2.1. The Original PCNN Model

PCNN is a two-dimensional neural network. Each neuron in the processing layer is directly mapped to a corresponding pixel in the image and generates the pulse output from a set of neighboring neurons. Generally, a model of a pulse-coupled neuron, illustrated in Figure 1, mainly consists of three parts as follows: (1) the input part, which takes the acceptance of external stimulus as well as the pulse output from neighboring neurons; (2) the coupled part, which increases the internal activity of neuron by coupling the linking part and feeding part; (3) the pulse output, which generates a pulse via comparison.

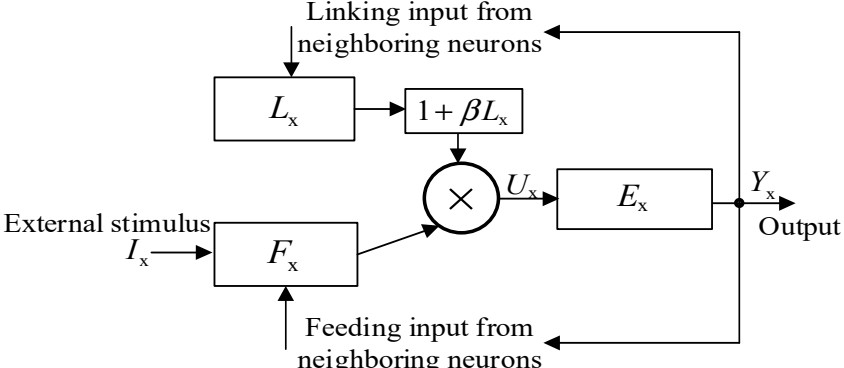

**Figure 1.** Schematic representation of a general pulse-coupled neuron.

In contrast to artificial neural networks, PCNN can be regarded as a third-generation neural network without any training. Its fundamental idea is to obtain the result by iterating the following equations.

$$F_x(n) = e^{-\alpha_F} F_x(n-1) + I_x + V_F \sum_{y \in N_x} M_{x,y} Y_y(n-1) \tag{1}$$

$$L_x(n) = e^{-\alpha_L} L_x(n-1) + V_L \sum_{y \in N_x} W_{x,y} Y_y(n-1) \tag{2}$$

$$U_x(n) = F_x(n) \cdot [1 + \beta L_x(n)] \tag{3}$$

$$E_x(n) = e^{-\alpha_E} E_x(n-1) + V_E Y_x(n-1) \tag{4}$$

$$Y_x(n) = \begin{cases} 1 & U_x(n) > E_x(n) \\ 0 & otherwise \end{cases} \tag{5}$$

Here, *F* and *L*, respectively, are the feeding and linking inputs, in which the neuron receives input from external stimulus *I* (e.g., intensity), and the pulse output *Y* is generated by neighboring neurons $N_y$. Basically, the neuron accumulates the internal activity *U* until it exceeds its inner threshold *E*. Once the neuron fires or generates a pulse ($Y_x = 1$), the result will make the threshold change, as well as the linking and feeding input and neighboring neurons at the next iteration. Through PCNN iteration, it will produce a sequence of binary images that can be utilized for image processing, such as segmentation, and feature extraction.

It is worth noting that the original PCNN model involves many parameters that can be altered to adjust the behavior of neurons. Among these parameters, some of them have proved to have a significant effect on image segmentation [2,17], especially the threshold constant $\alpha_E$ and linking coefficient $\beta$. Besides, the weight *W/M* also plays important role in connecting the neighboring neurons and transmitting the pulse to each other. For other parameters, three amplitude constants $V_F$, $V_L$ and $V_E$, cause the effect of the pulse output *Y*, while the decay constants $\alpha_F$ and $\alpha_L$ generate the dynamics of neuronal activity.

### 2.2. Modification of PCNN Model

In this part, a modified version of PCNN will be described, and the meaning of modified parts for subsequent parameter settings are given.

In the modified model, the major changes lie in the feeding input and linking input, neural threshold, as seen in Figure 2. What should be stressed is that some significant characteristics inherent in the original PCNN are maintained for image segmentation, such as synchronous pulse.

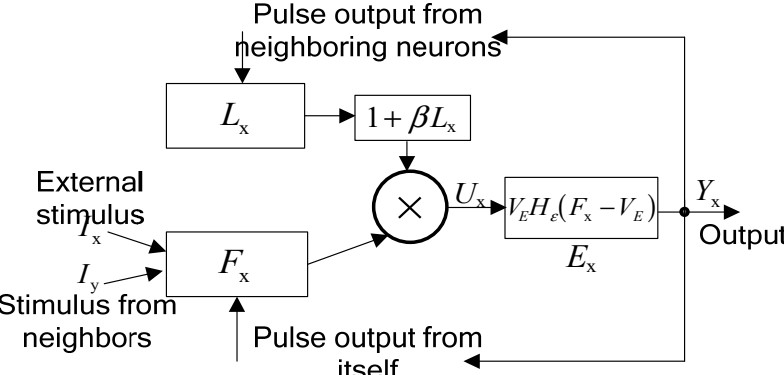

**Figure 2.** Schematic representation of a modified pulse-coupled neuron.

Let each neuron $x$ have an identical model, where $x \in \Omega$ mapped to the image domain. For a given neuron $x$, the equations of a single iterative procedure can be listed as follows:

$$F_x(n) = \sum_{y \in N_x} M_{x,y} I_y \tag{6}$$

$$L_x(n) = \sum_{y \in N_x} W_{x,y} Y_y(n-1) \tag{7}$$

$$U_x(n) = F_x(n) \cdot [1 + \beta L_x(n)] \tag{8}$$

$$E_x(n) = V_E \cdot H_E(F_x - V_E) \tag{9}$$

$$Y_x(n) = \begin{cases} 1 & U_x(n) > E_x(n) \\ 0 & otherwise \end{cases} \tag{10}$$

where all the notations have the same meaning as mentioned in Equations (1)–(5), except the notations in Equation (9). Here, $H_E$ denotes the Heaviside function, and is defined as

$$H_E(z) = \begin{cases} 0 & z \leq 0 \\ 1 & z > 0 \end{cases} \tag{11}$$

The move to these changes abandons some parts of the biological mechanism inherent in PCNN, but it allows for far greater control of the model behavior, making it worthwhile to promote the potential performance of segmentation.

## 3. Optimized Parameter Setting Method for PCNN

### 3.1. The Maximum Thresholding Rule

The maximum similarity thresholding (MST) rule is a kind of thresholding method, which measures the distance under the transformation space according to the result of the segmentation at different thresholds, and its representation is as follows

$$t^* = \underset{t}{\mathrm{argmax}}\{S(T(\chi), H(\gamma(t)))\} \tag{12}$$

where $T$ and $H$ are, respectively, transformation functions; $\chi$ denotes the original image; $\gamma(t)$ is the segmentation result under the threshold $t$; $S(\chi, \gamma(t))$ is the similarity measurement function, which is defined as follows:

$$S(\chi, \gamma(t)) = \frac{\sum_{i=1}^{N} (x_i - \mu_\chi)(\omega_i - \mu_\gamma)}{\sqrt{\sum_{i=1}^{N} (x_i - \mu_\chi)^2 \sum_{i=1}^{N} (\omega_i - \mu_\gamma)^2}} \tag{13}$$

where $x_i$ and $\omega_i$ correspond to the $i$-th pixel value in the image $\chi$ and image $\gamma(t)$, respectively; $\mu_\chi$ and $\mu_\gamma$ represent the mean of image $\chi$ and image $\gamma(t)$, and $N$ represents the number of pixels. Notably, the larger the $S$ value is, the more similar the images $\chi$ and image $\gamma(t)$ is.

Additionally, according to Equation (12), the segmentation result $\gamma$ is usually a binary image generated by the PCNN model. The similarity can be calculated directly with the image itself without $H$ transformation, the threshold value can be written by

$$t^* = \underset{t}{\text{argmax}}\{S(T(\chi), \gamma(t))\} \tag{14}$$

To find the optimal threshold, the optimal evaluation can be made by using images $\chi$ and $\gamma$. By introducing the boundary information into the transformation functions $H$ and $T$ in Equation (12), the final segmentation result of the threshold value is associated with the region boundary, thus laying a foundation for the complete extraction of the region. Without loss generation, the transformation functions $T$ can be set as follows.

$$T(\chi) = \|\nabla G(x, y;\ \sigma) * \chi\| \tag{15}$$

where $x$, $y$ denotes the position in the image; * denotes the convolution sign; $\nabla$ is the gradient operator; $\sigma$ denotes the scale value; $G$ is the Gaussian function as follows

$$G(x, y;\ \sigma) = \frac{1}{\sqrt{2\pi}\sigma} e^{-(x^2+y^2)/2\sigma^2} \tag{16}$$

In order to find the best value of $t$ from Equation (14), the transformation function $H$ usually adopts the boundary between the binary image of the target and the background produced after the segmentation. Here, the morphological operator for rapid processing is adopted, i.e.,

$$H(A) = A - (A \ominus B) \tag{17}$$

where $A$ is a thresholding result; $B$ is the operator from four-neighbor; $\ominus$ demotes the erode operator.

### 3.2. Linking Coefficient $\beta$

The linking coefficient $\beta$ inherent in PCNN is used to force the neighboring neurons with brightness similarity to pulse together. Inspired by the clustering method, the choice of designing the linking coefficient in this study is defined as follows

$$\begin{aligned} \min_{\beta}\quad & f(\beta) = \sum_{x \in X} \mu(S_x)(F_x - E_x)^2 \\ s.t.\quad & 0 < \beta < \xi \end{aligned} \tag{18}$$

where $\mu(\cdot)$ denotes the measure of the similarity, and the value of $\beta$ is not apparent in the cost function but is implied in the value of $S_x$ as follows

$$S_x = \begin{cases} F_x(1 + \beta L_x(n)) & U_x(n) > E_x(n) \\ F_x & otherwise \end{cases}, \quad x \in X \tag{19}$$

where $X$ is a set of eight-neighboring neurons around the boundary of the pulse output region.

In this work, to partition pixels into the background and object, the membership function associated with the distance to the obtained center of the object (background) is built as follows

$$\mu_o(g) = \begin{cases} 0 & g \leq a \\ \frac{1}{\left(1 + (g-b)^2/(g-a)^2\right)} & a < g \leq b \\ 1 & g > b \end{cases} \tag{20}$$

where $g$ is a variable in the intensity domain, $a$ is set as $m_2(n)$, and $b$ is set as $S_x$ at the $n$-th iteration. The intensity range $[a, b]$ is usually considered a fuzzy interval. From Equation (20), a set of pixels in $X$ can then transform from the intensity domain into the fuzzy domain, assigning a large membership value to the pixel whose intensity is close to the center.

To obtain the value of $\beta$, the golden section method was used to find the optimal solution, and a sufficiently large value $\xi = 2$ is set initially at each PCNN iteration.

In summary, the parameter setting of the modified PCNN model is associated with the characteristic of the synchronous pulse. The neighboring neurons surrounding the fired neurons are linked together by setting the linking coefficient $\beta$ and generate the pulse according to the maximum thresholding rule. There is a complete algorithm description in the following (see Algorithm 1).

| **Algorithm 1**: our PCNN model for image segmentation |
| --- |

| | |
| --- | --- |
| 1 Input | – test images; <br> – PCNN parameter M/W in literature [2] <br> – the initial PCNN neural threshold E whose value is the highest intensity of the image, and $\xi = 2$. |
| | **repeat** |
| 2 PCNN iteration | Compute the parameter $\beta$ in Equation (18); <br> Calculate $F$, $L$, and $U$ through iteration, as seen in Equations (6)–(8); <br> Update neural threshold $E$ in Equations (9) and (12); <br> Update $Y$ by Equation (10). |
| | **until** the pulse region Y does not change any more |
| 3 Output | –the marked region R as the final segmentation result |

## 4. Experimental Results

In this section, some experiments on real-world images (see Figure 3) are performed to assess the effectiveness of the proposed method. The results provided by the proposed method were then compared with the following segmentation methods, including Otsu's (OTSU) [31], Maximum similarity thresholding (MST) [32], Fuzzy c-means (FCM) [33], Meanshift clustering method [34]. In the experiments, without being otherwise specified, for a fair comparison, set $d = 1$, $\delta\beta = 0.01$, $\delta\mu_{max} = 0.15$, and $\beta_{max} = 1$ for Wei's method. In our model, the pulse output $Y(0)$ was initialized in which the neurons have the highest feeding input at each level. All the algorithms were implemented in Matlab 7.10 (2010a, MathWorks, Natick, America) and performed on a computer with Intel(R) Core (TM) 2.11 GHz i7 CPU RAM 16G, and Windows 64-bit operating system.

Figures 4–8 illustrate the segmentation results of OTSU, MST, FCM, mean shift clustering algorithm, and the proposed method, respectively. It is difficult to find that the threshold value obtained by the OTSU method can separate the background and the target. The MST algorithm adds gradient information from the background class and the target class to determine the optimal threshold, as shown in Figure 9. Generally, the edges of the target are clear to separate the target from the background. However, several maximum values of similarity occur during the threshold traversal, and the optimal threshold does not exist in the high peak value of maximum similarity, as shown in Figure 10. The threshold is listed as shown in Table 1. It can be found that the thresholds obtained by OSTU and MST methods are relatively low, so the segmentation results cannot effectively extract the target well.

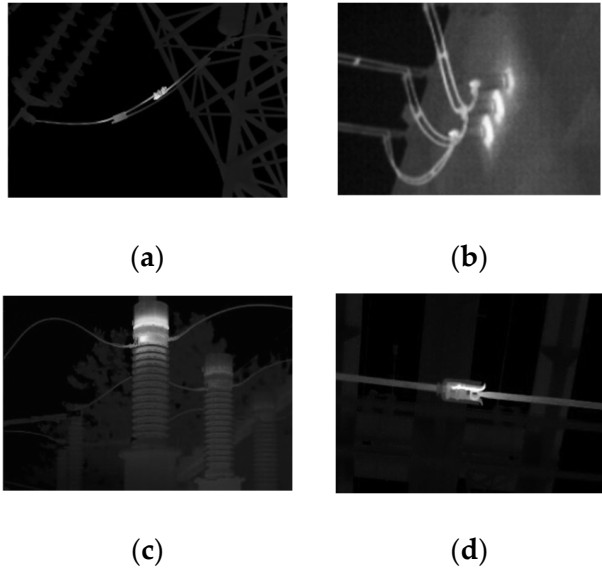

**Figure 3.** Real-world infrared images: (**a**) image 1; (**b**) image 2; (**c**) image 3; (**d**) image 4.

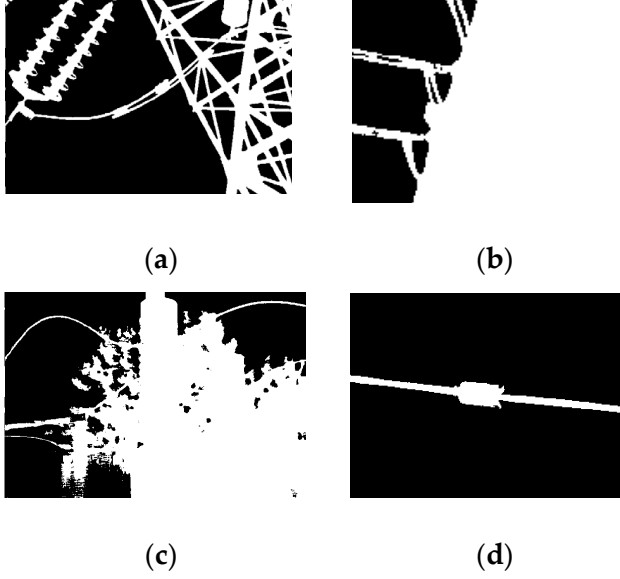

**Figure 4.** Segmentation results of OTSU method: (**a**) image 1; (**b**) image 2; (**c**) image 3; (**d**) image 4.

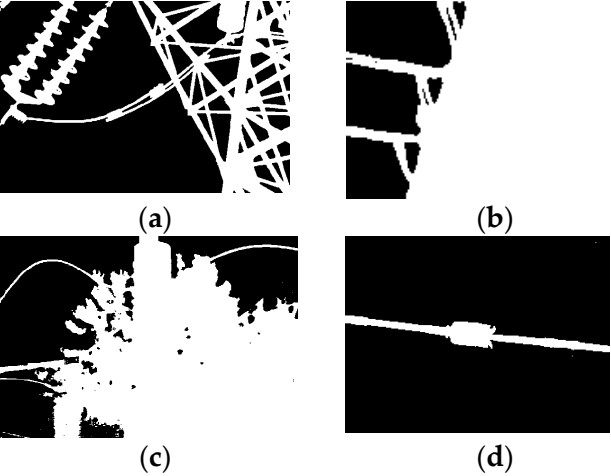

**Figure 5.** Segmentation results of MST method: (**a**) image 1; (**b**) image 2; (**c**) image 3; (**d**) image 4.

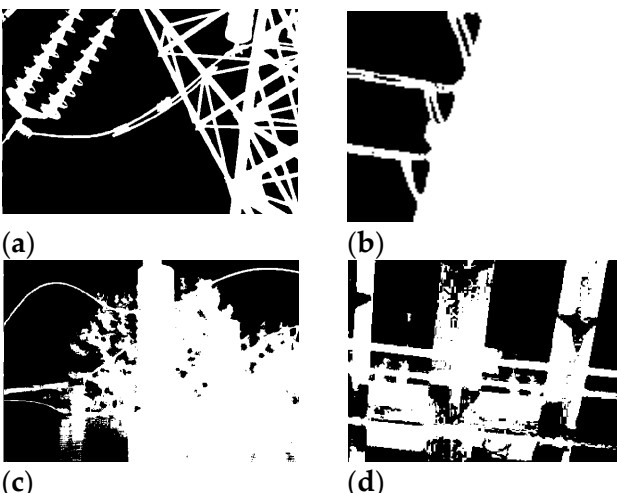

**Figure 6.** Segmentation results of FCM method: (**a**) image 1; (**b**) image 2; (**c**) image 3; (**d**) image 4.

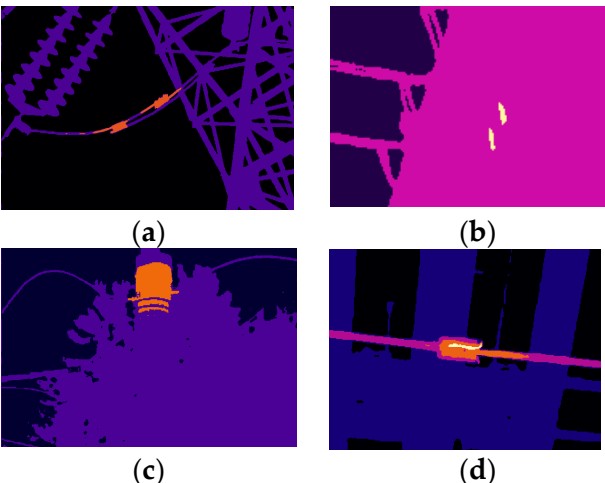

**Figure 7.** Segmentation results of Mean-shift method: (**a**) image 1; (**b**) image 2; (**c**) image 3; (**d**) image 4.

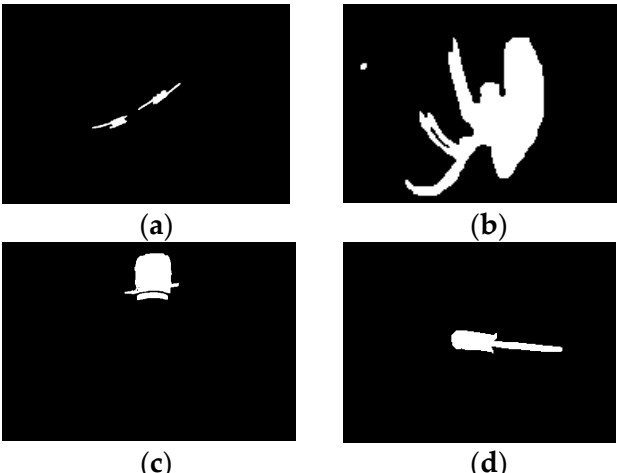

**Figure 8.** Segmentation results of the proposed method: (**a**) image 1; (**b**) image 2; (**c**) image 3; (**d**) image 4.

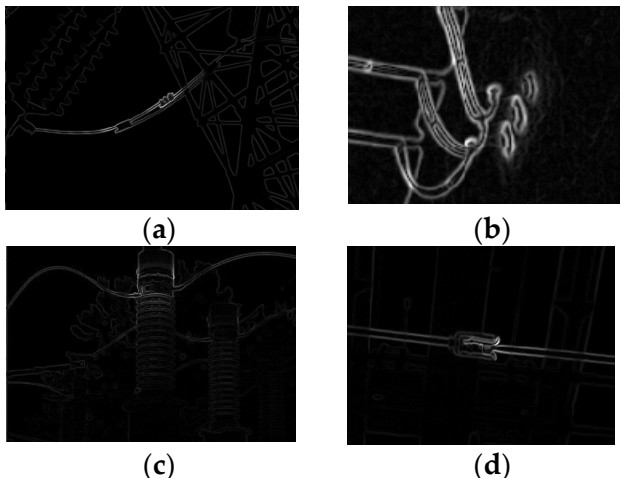

**Figure 9.** The gradient result of infrared images: (**a**) image 1; (**b**) image 2; (**c**) image 3; (**d**) image 4.

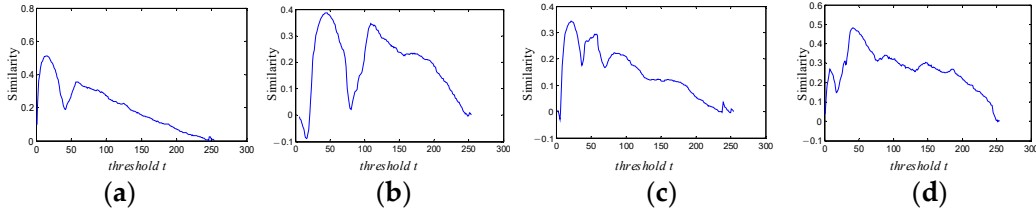

**Figure 10.** Similarity of MST method: (**a**) image 1; (**b**) image 2; (**c**) image 3; (**d**) image 4.

**Table 1.** Thresholding of MST and OTSU.

|      | Image 1 | Image 2 | Image 3 | Image 4 |
|------|---------|---------|---------|---------|
| OTSU | 21      | 59      | 31      | 57      |
| MST  | 15      | 45      | 22      | 41      |

Compared with the threshold method, the FCM method classifies pixels based on the principle of maximum membership degree. However, the FCM method may classify the intensity of the pixels without the consideration of target distribution, because the target often appears with high intensity in the image. The final means of classes are given in Table 2. From the segmentation results shown in Figure 8, the mean values of the background and the target (i.e., $v_1$ and $v_2$) obtained by the proposed method seem close to the actual value, although some pixels are a misclassification. By comparison, the mean values obtained by the FCM method are not better than that of the proposed method. Some of them are biased from the actual centers.

**Table 2.** Mean value of final results.

|                 | Mean Value | Image 1  | Image 2  | Image 3  | Image 4  |
|-----------------|------------|----------|----------|----------|----------|
| FCM             | $v_1$      | 1.0603   | 21.05    | 1.01730  | 5.7834   |
|                 | $v_2$      | 41.3490  | 96.8447  | 53.8848  | 31.1222  |
| Proposed method | $v_1$      | 14.1560  | 66.2512  | 30.4860  | 14.9498  |
|                 | $v_2$      | 132.1046 | 132.1663 | 145.4363 | 129.3911 |

The Meanshift algorithm is based on the maximum clustering of probability density, and therefore, it is superior for pixel clustering. It can be found in Figure 7 that the whole image is divided into multiple regions, eventually. However, for this clustering method with great probability density characteristics, it is also closely related to the grayscale distribution of regional pixels. For example, in the second image segmentation results, part of the high-brightness area of the transmission line is classified into other areas. Moreover,

for the fourth image, the presence of multiple cluster areas around the failure area occurred due to the inhomogeneity of the target.

In the proposed method, the segmentation performance is associated with the maximum similarity thresholding during PCNN iteration. The results are shown in Figure 8. From the similarity evaluation of the segmentation results in Table 3, it can be found that the evaluation value of the proposed method is lower than the global similarity, but the obtained results are better than the MST algorithm. Additionally, combined with the membership assignment method for adjusting the linking coefficient, the region extraction effect had significant improvement.

**Table 3.** Evaluation of similarity.

|  | Image 1 | Image 2 | Image 3 | Image 4 |
|---|---|---|---|---|
| MST | 0.5120 | 0.3872 | 0.3433 | 0.4815 |
| Proposed | 0.3430 | 0.2419 | 0.2445 | 0.3515 |

Table 4 lists the contrast in the time complexity of the mentioned methods. The OTSU method is the least time-consuming, and the MST methods as threshold segmentation have low time consumption. By contrast, FCM and Meanshift as clustering algorithms need the iterative clustering of image pixels, so the time complexity is higher than the thresholding method. The proposed method is partly limited by similarity threshold calculation and neighborhood membership assignment and iterative clustering, so the time complexity increases compared to MST, but it is lower compared to clustering methods such as FCM and Meanshift algorithm,

**Table 4.** Evaluation of time complexity.

|  | Image 1 | Image 2 | Image 3 | Image 4 |
|---|---|---|---|---|
| OTSU | 0.1402 | 0.0022 | 0.1076 | 0.0042 |
| MST | 3.7802 | 0.2007 | 3.4117 | 0.9979 |
| FCM | 5.6357 | 0.9010 | 13.6103 | 7.7979 |
| Meanshift | 5.5414 | 0.2870 | 7.5225 | 1.1907 |
| Proposed | 3.2133 | 0.1252 | 3.2280 | 0.3673 |

In the near future, the algorithm will be applied to the safety production risk control platform. Combined with the operation control module of 3D information, it is very convenient to manage and control the electric power equipment fault. Moreover, the early warning of security risks can be predicted, thus improving the power grid engineering safety risk identification and operation control ability.

## 5. Conclusions

In this paper, an effective strategy for the diagnosis of power equipment in the transmission line is proposed which is based on a modified PCNN. The parameters of the PCNN model can interact with each other via the characteristic of synchronous pulse and the maximum similarity thresholding rule, allowing the model to be self-organized to segment images. By incorporating the local clustering into the framework of the linking coefficient, our model can segment the image with high performance. Finally, in order to assess the effectiveness of the proposed model, the experiments on the real-world images are tested, and the results yielded by the proposed method have more competition than that yielded by some existing segmentation methods and have demonstrated the advantage of our model in terms of strategy to image segmentation. However, the critical issues that should further be considered consist of the following aspects: (1) the drift of the maximum threshold occurs when the intensity-inhomogeneity appears in the infrared target, making the performance of PCNN segmentation become worse; (2) During the calculation of the linking coefficient, the obtained small value often make more iterations. In the near future, a great effort will

be made to adjust the threshold by considering the intensity-inhomogeneity and finding a better rule for the selection of the linking coefficient. Furthermore, by combining with the three-dimensional information of the operation control module, it can promote the ability of safety risk identification and operation control of power grid engineering in the inspection of a transmission line.

**Author Contributions:** Y.T. provided the idea of work, J.T. implemented the idea by MATLAB, X.Z. organized the framework of this manuscript, C.C. set up the experiment platform, Z.C. collected the experiment data, and Z.H. conducted the calculation. All authors have read and agreed to the published version of the manuscript.

**Funding:** This research was funded by the research on security Risk identification and Operation Control technology of Power grid Engineering based on 3D spatial information fusion, grant number 5700-202233182A-1-1-ZN, and Supported by the Science and Technology Project of Headquarters of State Grid Corporation of China.

**Data Availability Statement:** The raw data supporting the conclusion of this article will be made available by the authors, without undue reservation.

**Conflicts of Interest:** The authors declare no conflict of interest.

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
