# Peer review of "Transmission Line Equipment Infrared Diagnosis Using an Improved Pulse-Coupled Neural Network"

_sustainability, doi:10.3390/su15010639_

Round 1
Reviewer 1 Report
According to my opinion, this paper is interesting. The paper presents a novel idea and therefore is appropriate for an International Journal publication. There is an appropriate theory, logical and numerical concept that is necessary for originality of presented paper.
In this paper, a novel strategy for image segmentation is presented by a modified version of pulse-coupled neural network (PCNN). The fundamental idea of this method is to merge a similarity thresholding method. Then, a new relationship between the parameters and the available information like the pulse output and image characteristics is built to enable the model to extract the target. The main contributions of this paper are related to two important methodological issues. The first one is related to a similarity thresholding method. The second contribution concerns the critical problem of the iterative mechanism inherent in PCNN for infrared image segmentation.
To validate a novel strategy for image segmentation, in the experiments the real-world images are tested. The results yielded by the proposed method have more competition than that yielded by some existing segmentation methods and have demonstrated the advantage of proposed model in term of strategy to image segmentation.
Author Response
Response to Reviewer 1 Comments
According to my opinion, this paper is interesting. The paper presents a novel idea and therefore is appropriate for an International Journal publication. There is an appropriate theory, logical and numerical concept that is necessary for originality of presented paper.
In this paper, a novel strategy for image segmentation is presented by a modified version of pulse-coupled neural network (PCNN). The fundamental idea of this method is to merge a similarity thresholding method. Then, a new relationship between the parameters and the available information like the pulse output and image characteristics is built to enable the model to extract the target. The main contributions of this paper are related to two important methodological issues. The first one is related to a similarity thresholding method. The second contribution concerns the critical problem of the iterative mechanism inherent in PCNN for infrared image segmentation.
To validate a novel strategy for image segmentation, in the experiments the real-world images are tested. The results yielded by the proposed method have more competition than that yielded by some existing segmentation methods and have demonstrated the advantage of proposed model in term of strategy to image segmentation.
Response : Many thanks, we are feel so warm for your comment.

Reviewer 2 Report
You claim your novelty that PCNN (modified) may i know what exactly modifies.
a methodology flow chart may be added so it has more values
you chosen transmission of which voltage (mentioned voltage level)
Author Response
Response to Reviewer 2 Comments
Point 1: You claim your novelty that PCNN (modified) may i know what exactly modifies.
Response 1: Thanks for your comments. The main contributions of this paper are related to two important methodological issues. The first one is related to a maximum thresholding rule to determine the threshold of the modified PCNN model. The second contribution concerns the regulation of the linking coefficient, activating similar neurons as much as possible to make the segmentation performance better. We have added it in the last two paragraph of Sect. 1
Point 2: a methodology flow chart may be added so it has more values.
Response 2: Thanks for your suggestion. For more understanding, we have added a complete algorithm description for the modified PCNN model, as listed in the last paragraph of Sect. 3.
Point 3: you chosen transmission of which voltage (mentioned voltage level)
Response 3: Thanks for your comment. In this work, the images used in the test are from 110kv voltage level.
Thank you very much for your great efforts on our manuscript. We also appreciate you for the valuable suggestions and questions.

Reviewer 3 Report
The authors have prepared a very interesting article.
- In the introduction, the contribution of the work is clearly highlighted and the research is compared in detail with the relevant literature. However, I think the Introduction should be numbered as 1 instead of 0, next Section as 2, ect.
- The article is well structured and the methods are clearly explained.
- The developed method was confirmed experimentally on real examples from practice.
- The results are presented with descriptions, figures and tables, but with a few errors that need to be fixed:
- add the full name of the methods OTSU, MST, FCM, at the first mention,
- clarify the meaning of the values in tables 1 and 2 and the labels v1, v2 (it is certainly clear to the authors, but readers may need help to interpret the significance of these values, whether it is better that it is smaller, larger, close to a certain amount, or...)
- in the text (lines 241,242) the segmentation performance of the proposed method is mentioned, it should be added that this is shown in Figure 8, while the comments about Figures 9 and 10 are completely missing.
- in line 243, Table 3 showing the similarity evaluation is mentioned, but it is missing in the paper, on the next page is Table 3 showing the evaluation of time complex, should it have been Table 4?
- The conclusion summarizes well the main goals and contributions of the work, and indicates the further direction of the research and the possibilities of improving the method. I suggest using the passive voice (instead of "we") in the conclusion also, that is, the expression proposed model instead of our model.
Author Response
Response to Reviewer 3 Comments
The authors have prepared a very interesting article.
Point 1: In the introduction, the contribution of the work is clearly highlighted and the research is compared in detail with the relevant literature. However, I think the Introduction should be numbered as 1 instead of 0, next Section as 2, etc.
Response 1: Thanks for your comment. we have checked it and modified the section number.
Point 2: The article is well structured and the methods are clearly explained.
Response 2: Thanks for your comment.
Point 3: The developed method was confirmed experimentally on real examples from practice
Response 3: Thanks for your comment.
Point 4: The results are presented with descriptions, figures and tables, but with a few errors that need to be fixed.
- add the full name of the methods OTSU, MST, FCM, at the first mention,
Response 4: Thanks for your comment.
--we have added the full name of the methods OTSU, MST, FCM, at the first mention.
- clarify the meaning of the values in tables 1 and 2 and the labels v1, v2 (it is certainly clear to the authors, but readers may need help to interpret the significance of these values, whether it is better that it is smaller, larger, close to a certain amount, or...)
Response 4: Thanks for your comment. Table 1 lists the thresholds obtained by OSTU and MST methods. They are relatively low, so the segmentation results can not effectively extract the target well. Table 2 lists the final means of classes. From the segmentation results, the mean values of the background and the target (i.e, v1 and v2) obtained by the proposed method seem close to the actual value, although some pixels are misclassification. By comparison, the mean values obtained by the FCM method are not better than that of the proposed method. Some of them are biased from the actual centers.
- in the text (lines 241,242) the segmentation performance of the proposed method is mentioned, it should be added that this is shown in Figure 8, while the comments about Figures 9 and 10 are completely missing.
Response 4: Thanks for your comment. we have added the description of Figure 9 and Figure 10. The MST algorithm adds gradient information from the background class and the target class to determine the optimal threshold, as shown in Figure 9. Generally, the edges of the target are clear to separate the target from the background. However, several maximum values of similarity occur during the threshold traversal, and the optimal threshold does not exist in the high peak value of maximum similarity, as shown in Figure 10.
- in line 243, Table 3 showing the similarity evaluation is mentioned, but it is missing in the paper, on the next page is Table 3 showing the evaluation of time complex, should it have been Table 4?
Response 4: Thanks for your comment. This is my mistake, and now we have added Table 3 showing the similarity evaluation.
- The conclusion summarizes well the main goals and contributions of the work, and indicates the further direction of the research and the possibilities of improving the method. I suggest using the passive voice (instead of "we") in the conclusion also, that is, the expression proposed model instead of our model.
Response 4: Thanks for your comment. We have modified it in the conclusion.
Thank you very much for your great efforts on our manuscript. We also appreciate you for the valuable suggestions and questions.

Reviewer 4 Report
My comments are as follows :
1- please explain main numerical results in Abstract
2- please determinates the research gaps of literature review,
3- what are the contributions and novelty of paper
4- wate are limits of work
5- please explain the future works
Author Response
Response to Reviewer 4 Comments
The authors have prepared a very interesting article.
Point 1: please explain main numerical results in Abstract.
Response 1: Thanks for your comments. In the revised maunscript, we have checked and explained it in Abastract.
Point 2: please determinates the research gaps of literature review.
Response 2: Thanks for your comment. In the revised manuscript, we collated the literature and focus on the work. In general, the performance of the PCNN model to image segmentation is heavily dependent on the values of network parameters. The intensity-inhomogeneity often occurs, which makes PCNN unsuitable for image segmentation according to the existing parameter setting methods. Besides, there are many model parameters, so it is not easy to build a reasonable setting method for promoting the performance of image segmentation. In the revised manuscript, we have added it in the last three paragraph in Sect. 1.
Point 3: what are the contributions and novelty of paper
Response 3: Thanks for your comment. The main contributions of this paper are related to two important methodological issues. The first one is related to a maximum thresholding rule to determine the threshold of the modified PCNN model. The second contribution concerns the regulation of the linking coefficient, activating similar neurons as much as possible to make the segmentation performance better. In the revised manuscript, we have added it in the last two paragraph in Sect. 1.
Point 4: wate are limits of work
Response 4: Thanks for your comment. In this manuscript, the critical issues that should be considered consist of the following aspects: (1) the drift of the maximum threshold occurs when the intensity-inhomogeneity appears in the infrared target, making the performance of PCNN segmentation become worse; (2) During the calculation of the linking coefficient, the obtained small value often make more iterations. We have added it to the conclusion.
Point 4: please explain the future works
Response 4: Thanks for your comment. In the near future, a great effort will be made to adjust the threshold by considering the intensity-inhomogeneity and finding a better rule for the selection of the linking coefficient. Furthermore, by combining with the three-dimensional information of the operation control module, it can promote the ability of safety risk identification and operation control of power grid engineering in the inspection of a transmission line. We have added it to the conclusion.
Thank you very much for your great efforts on our manuscript. We also appreciate you for the valuable suggestions and questions.

Round 2
Reviewer 4 Report
My comments are answered. Thanks